# Disseminated disease due to non-tuberculous mycobacteria in HIV positive patients: A retrospective case control study

**Nils Wetzstein**[1]*, **Ari Geil**[1], **Gerrit Kann**[1], **Annette Lehn**[2], **Gundolf Schüttfort**[1], **Johanna Kessel**[1], **Tobias M. Bingold**[1], **Claus P. Küpper-Tetzel**[1], **Annette Haberl**[1], **Christiana Graf**[3], **Maria J. G. T. Vehreschild**[1], **Christoph Stephan**[1], **Michael Hogardt**[4], **Thomas A. Wichelhaus**[4], **Timo Wolf**[1]

**1** Department of Internal Medicine, Infectious Diseases, University Hospital Frankfurt, Goethe University, Frankfurt am Main, Germany, **2** Institute of Biostatistics and Mathematical Modeling, Goethe University, Frankfurt am Main, Germany, **3** Department of Internal Medicine, Gastroenterology and Hepatology, University Hospital Frankfurt, Goethe University, Frankfurt am Main, Germany, **4** Institute of Medical Microbiology and Infection Control, University Hospital Frankfurt, Goethe University, Frankfurt am Main, Germany

* nils.wetzstein@kgu.de

**Data Availability Statement:** Individual patients' data cannot be shared publicly because of the European data protection regulation, as these data might be used to identify patients indirectly. In

## Abstract

### Introduction

Disseminated infection due to non-tuberculous mycobacteria has been a major factor of mortality and comorbidity in HIV patients. Until 2018, U.S. American guidelines have recommended antimycobacterial prophylaxis in patients with low CD4 cell counts, a practice that has not been adopted in Europe. This study aimed at examining the impact of disseminated NTM disease on clinical outcome in German HIV patients with a severe immunodeficiency.

### Materials and methods

In this retrospective case control study, HIV patients with disseminated NTM disease were identified by retrospective chart review and matched by their CD4 cell counts to HIV patients without NTM infection in a 1:1 alocation. Primary endpoints were mortality and time to first rehospitalisation. In addition, other opportunistic diseases, as well as antimycobacterial and antiretroviral treatments were examined.

### Results

Between 2006 and 2016, we identified 37 HIV patients with disseminated NTM disease. Most of them were suffering from infections due to *M. avium* complex (*n* = 31, 77.5%). Time to event analysis showed a non-significant trend to higher mortality in patients with disseminated NTM disease (p = 0.24). Rehospitalisation took place significantly earlier in patients with disseminated NTM infections (median 40.5 days vs. 109 days, p<0.0001).

addition, it was agreed with our local Ethics Committee that no primary data that might be used to identify patients are shared with third parties. Nevertheless, data are available from the Ethics Committee at University Hospital Frankfurt (contact via ethikkommission@kgu.de) for researchers who meet the criteria for access to confidential data. All other relevant data are within the manuscript.

**Funding:** The author(s) received no specific funding for this work.

**Competing interests:** Nils Wetzstein, Ari Geil, Gerrit Kann, Annette Lehn, Johanna Kessel, Tobias M. Bingold, Claus P. Küpper-Tetzel, Christiana Graf and Thomas A. Wichelhaus have nothing to disclose. Gundolf Schüttfort received funding from Gilead Sciences and speaker fees from ViiV Healthcare, Bristol Meyer Squibb, MSD and Hormosan for participation in Advisory Boards, Data Safety and Monitoring Boards and for preparation of educational materials and lecturing fees, all outside the submitted work. Annette Haberl received speaker fees from Gilead Sciences, Janssen-Cilag and MSD and also participated in MSD advisory boards. Support of congress participation was provided by Gilead Sciences and Janssen-Cilag. Maria JGT Vehreschild has received research grants from 3M, Astellas Pharma, Biontech, DaVolterra, Evonik, Gilead Sciences, Glycom, Immunic, MaaT Pharma, Merck/MSD, Organobalance, Seres Therapeutics, Takeda Pharmaceutical, has received speaker fees from Astellas Pharma, Basilea, Gilead Sciences, Merck/MSD, Organobalance, Pfizer and been a consultant to Alb Fils Kliniken GmbH, Arderypharm, Astellas Pharma, DaVolterra, Farmak International Holding GmbH, Ferring, Immunic AG, MaaT Pharma, Merck/MSD, SocraTec R&D GmbH, all outside the submitted work. Christoph Stephan has received research grants from Gilead Sciences, Janssen, MSD, ViiV-Healthcare and has received fees for scientific advice from Merck/MSD, and Theratechnologies, all outside the submitted work. Michael Hogardt received Honoraria for consulting (Chiesi GmbH), and lectures (Chiesi GmbH/Horizon Pharma/Thieme Publisher Group), all outside the submitted work. Timo Wolf received fees for lectures, consultancies and travel grants from: Gilead, Merck Sharp Dome, Janssen, all outside the submitted work. This does not alter our adherence to PLOS ONE policies on sharing data and materials.

**Abbreviations:** AIDS, acquired immune deficiency syndrome; ART, antiretroviral therapy; BMI, Body mass index; CCI, Charlson comorbidity index; CMV, cytomegalovirus; COPD, chronic obstructive pulmonary disease; EBV, Epstein-Barr

## Conclusion

In this retrospective case control study, we could demonstrate that mortality is not significantly higher in HIV patients with disseminated NTM disease in the ART era, but that they require specialised medical attention in the first months following discharge.

## Introduction

Infections due to non-tuberculous mycobacteria (NTM) have been observed in HIV patients since the advent of the AIDS pandemic in the 1980s [1]. The widespread administration of antiretroviral therapy (ART) has led to a decrease in morbidity and mortality due to opportunistic infections in HIV-positive patients and an increase in life expectancy [2,3]. Nevertheless, patients with low CD4 cell counts at first diagnosis are still suffering from a variety of opportunistic infections and/or AIDS related malignancies [4]. Among those, disseminated disease due to non-tuberculous mycobacteria is a major cause of morbidity [5].

Mainly, species of the *M. avium* complex (MAC, formerly called MAI–*M. avium-intracellulare-* complex) have been involved in disseminated infections in severely immunocompromised patients. Other NTM (as for example *M. genavense)*, have also been described [1,6,7]. In disseminated disease, NTM can be cultured from different specimens such as citrate or EDTA blood, organ biopsies, sputum, bronchoalveolar lavage fluid and others. This clinical entity has to be distinguished from mere pulmonary colonisation or NTM pulmonary disease (NTM-PD).

Until recently, primary prophylaxis against MAC disease was generally recommended in patients with HIV and CD4 cell counts below 50/μl in the United States [8]. Since 2018, only patients who cannot begin ART immediately should receive mycobacterial prophylaxis [9]. In Europe, NTM prophylaxis has never been adopted, as MAC infections have been described more rarely [10,11].

Disseminated NTM infections are now frequently diagnosed in relation to an immune-reconstitution-syndrome due to NTM (NTM-IRIS) causing severe symptoms such as multiple abscesses, lymphadenopathy and fever of unknown origin after the initiation of ART [12,13]. Those cases can be especially difficult to treat [12]. Whereas the recommended antimicrobial regimen for disseminated *M. avium complex* disease consists of a macrolide, ethambutol and a rifamycin, there is little data on the treatment of rare NTM like *M. genavense* or *M. xenopi* [1,9].

The aim of this study was to examine the impact of disseminated NTM disease on clinical outcome in HIV patients with a severe immunodeficiency in a healthcare environment without recommendation for antimycobacterial prophylaxis. In addition, we intended to identify clinical characteristics associated with NTM infections in HIV patients and a possible influence of antimycobacterial treatment on their clinical course.

## Materials and methods

Ethical consent was obtained under file number 473/16 from our local ethics committee at University Hospital Frankfurt. For the period between 2006 and 2016, all patients treated at our centre with coded HIV infection (ICD codes B20-B24), coded NTM infection (ICD code A31) and positive microbiological results for NTM were identified and doubles excluded. Patient data was then collected by retrospective chart review of our local patient management

Virus; EDTA, Ethylenediaminetetraacetic acid; HIV, Humane immunodeficiency virus; HSV, *Herpes simplex* virus; IQR, interquartile range; IRIS, immune reconstitution syndrome; ITP, idiopathic thrombocytopenia; IVDU, iv drug use; MAC, *Mycobacterium avium* complex; NNRTIs, non-nucleos(t)ide reverse transcriptase inhibitor; NRTI, nucleos(t)ide reverse transcriptase inhibitor; NTM, non-tuberculous mycobacteria; NTM-PD, NTM pulmonary disease; PCP, *Pneumocystis jirovecii* pneumonia; PCR, polymerase chain reaction; PNP, polyneuropathy; PPY, per patient year; SD, standard deviation; TB, tuberculosis; VZV, *Varicella zoster* virus.

system (ORBIS, Dedalus Health Care Systems Group, Bonn Germany). Medical records were accessed between October 2018 and December 2020. As this is a retrospective study, no informed consent was required. Before further statistical analysis, all patient data were anonymised.

Patients were grouped into two clinical cohorts: patients with HIV infection and diagnosed disseminated NTM infection (*disseminated NTM disease*) and patients with HIV infection but without NTM infection (*controls*). Disseminated NTM infection was defined by positive culture or PCR result for NTM in either a blood specimen (EDTA or citrate culture) or more than one specimen from different body sites. Patients, in which NTM were cultured from respiratory specimens only, were excluded (*NTM-PD or colonisation*).

Case control matching was then conducted using the *MatchIt* package in R [14,15]. In a first step, CD4 cell-counts at the time of first diagnosis of NTM in disseminated NTM disease and the nadir (lowest CD4 cell count measured during the observation period) in controls were recorded. Then, patients were matched with regard to CD4 cell count with a 1:1 ratio, while only controls were discarded.

For the resulting groups the following outcome variables were recorded: mycobacterial species as identified by our clinical microbiology laboratory (case group, disseminated NTM disease), fatal events, reason of death and possible time to death, hospitalisation rate and time to first rehospitalisation, total observation time, Charlson Comorbidity Index (CCI) [16,17], behavioral risk factors (such as smoking, intravenous drug use–IVDU—and alcohol abuse), opportunistic infections and AIDS related malignancies. NTM treatment was recorded and divided into the following antimicrobial groups: macrolides, rifamycin, ethambutol, fluoroquinolones and others. HIV treatment was grouped into NRTIs, NNRTIs, protease inhibitors, integrase inhibitors, fusion inhibitor and CC motif chemokine receptor 5-(CCR5) inhibitor.

In patients, in whom MAC isolates were still available, isolates were cultivated as described previously [18] and species identification was reevaluated using the NTM DR genotype line probe as recommended by the manufacturer [19]. MAC could thus be specified into *M. avium*, *M. chimaera* and *M. intracellulare*.

All statistical analyses were performed in R v. 3.4.4, Someone to Lean On"[14] and LibreOffice (The Document Foundation). All graphs were drawn using the *ggplot* package [20]. Continuous variables are presented as median and interquartile range (IQR) for non-normally distributed data or mean and standard deviation (SD) for normally distributed data. Frequencies for categorical variables are presented as number and percentage. The Shapiro-Wilk-Test was used to test for normality. For non-normally distributed data, significance tests were performed using the Wilcoxon-Mann-Whitney test, for normally distributed data the student's t-test. Significant differences of categorical variables were evaluated using the Fisher exact test or chi-squared test. All tests were performed two-sided at a significance level of alpha = 0.05.

Kaplan-Meier method with log-rank test was used for time to event analysis using the *survival* and *survminer* packages in R [21,22].

## Results

### Patients and matching process

In total, we could identify 398 patients with an HIV infection available for further analysis (Fig 1). Forty of those were excluded, because NTM were cultivated only from respiratory specimens (*NTM-PD or colonisation*). Thirty-seven patients met our definition of disseminated disease due to NTM and formed the case group (*disseminated NTM disease*). Matching excluded

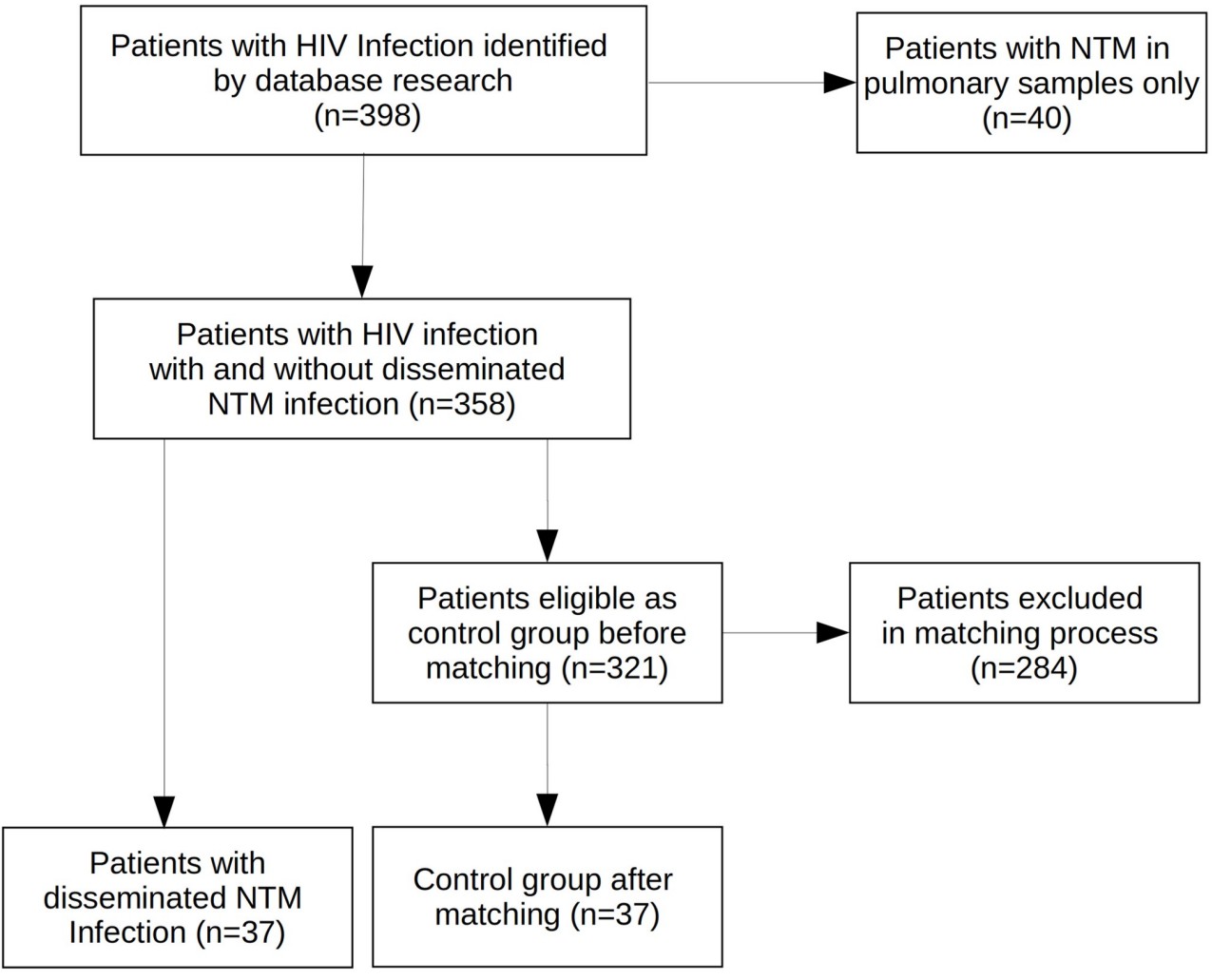

**Fig 1. Flowchart of patient inclusion and exclusion.**

284 patients, delivering a control group of 37 patients, as well (*controls*). Median CD4 cell count at first diagnosis was 5/μl (IQR = 22/μl) in patients with disseminated NTM disease, 198/μl (IQR = 273/μl) in controls before matching and 11/μl (IQR = 26/μl) in controls after matching (Fig 2). There was no significant difference in CD4 cell count between the case and the control group after the matching process (p = 0.33).

## General characteristics and mycobacterial species distribution

MAC was the most frequent mycobacterial species group causing disseminated infection (*n* = 31, 83.8%). Other mycobacterial species were: *M. genavense* (*n* = 2, 5.4%), *M. kansasii*, *M. celatum*, *M. simiae* and *M. mycogenicum* (*n* = 1, 2.7% each) (Table 1). In 26 patients, MAC isolates were available. Among those, we identified *M. avium* only, but no representatives of *M. chimaera*, or *M. intracellulare* by the line probe assay.

There were no significant differences between the case and control group regarding age, BMI, CCI, ethnicity and gender. Viral load was significantly higher and the observation period significantly longer in patients of the control group.

# CD4 cells before and after matching

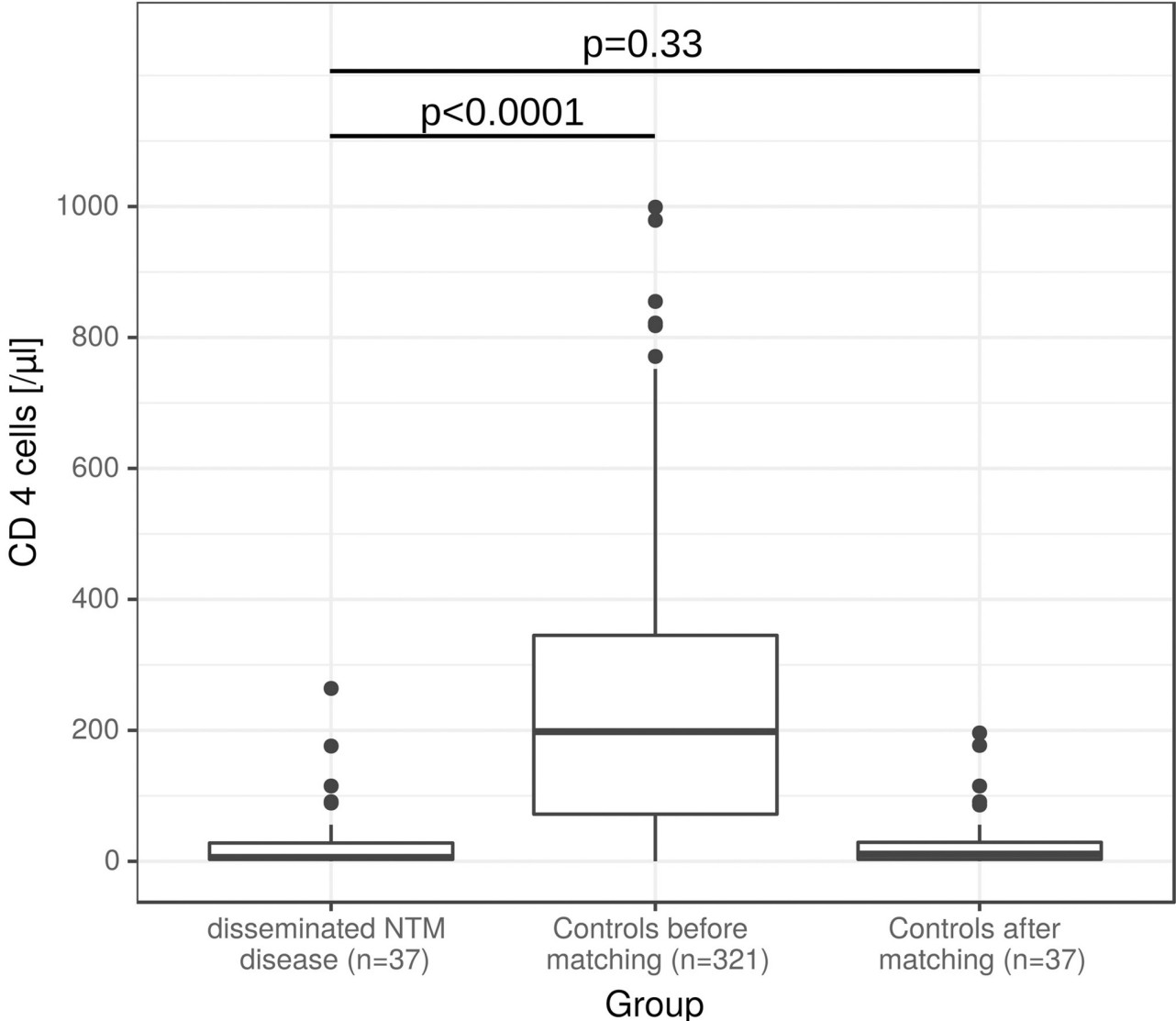

**Fig 2. Boxplot of CD4 cell counts [/µl] in patients with disseminated NTM disease (*n* = 37), controls before matching (*n* = 321) and controls after matching (*n* = 37).**

## Opportunistic infections and other clinical attributes

All opportunistic infections were equally distributed among patients with disseminated NTM disease and the control group except for infections due to cytomegalovirus (CMV) (p<0.01), and a trend to higher frequency of *Pneumocystis jirovecii* pneumonia (PCP) in the case group (p = 0.07) (Tables 2 and 3). Fever, night sweat and weight loss were significantly more frequent in patients suffering from disseminated NTM infections (75.0% vs 32.4%, p<0.001; 54.2% vs 16.2%, p<0.01; 85.3% vs 27%, p<0.001). Blood samples for mycobacterial culture were drawn significantly more often in patients with disseminated NTM disease than in controls (89.2% vs. 46.0%, p<0.001). In the case group, there was a trend to more documented cases of IRIS (18.9% vs. 2.7%, p = 0.06) and wasting syndrome (21.6% vs.5.4%. p = 0.09).

**Table 1. General characteristics and mycobacterial species in patients with disseminated NTM disease and controls.** CD4-cell counts refer to baseline in patients with disseminated NTM disease and the nadir in controls.

| | Disseminated NTM disease (n = 37) | | Controls (n = 37) | | p value |
|---|---|---|---|---|---|
| **Categorical variables** | **n** | **[%]** | **n** | **[%]** | |
| Gender | | | | | |
| Male | 25 | 67.6 | 28 | 75.7 | 0.61 |
| Female | 12 | 32.4 | 9 | 24.3 | |
| Ethnicity | | | | | |
| Caucasian | 27 | 73.0 | 25 | 67.6 | 0.49 |
| African | 8 | 21.6 | 8 | 21.6 | |
| Hispanic | 0 | 0 | 0 | 0 | |
| Asian | 1 | 2.7 | 4 | 10.8 | |
| Middle East | 1 | 2.7 | | 0 | |
| Mycobacterium | | | | | |
| *MAC* | 31 | 83.8 | NA | NA | |
| *M. kansasii* | 1 | 2.7 | NA | NA | |
| *M. celatum* | 1 | 2.7 | NA | NA | |
| *M. simiae* | 1 | 2.7 | NA | NA | |
| *M. genavense* | 2 | 5.4 | NA | NA | |
| *M. mycogenicum* | 1 | 2.7 | NA | NA | |
| **Continuous variables** | **Median** | **IQR** | **Median** | **IQR** | |
| Age [years] | 39.2 | 17.8 | 44.0 | 12.0 | 0.12 |
| CD4-cells [/μl] | 6.0 | 29.6 | 11.0 | 26 | 0.33 |
| HIV viral load [cp/ml] | 35200.0 | 1481500.0 | 52000.0 | 112370.0 | **< 0.001** |
| BMI [kg/m$^2$] | 19.0 | 4.9 | 19.2 | 20.4 | 0.68 |
| CCI | 7 | 2 | 6.0 | 2.0 | 0.09 |
| Total observation time [days] | 957 | 1915 | 1992 | 1604 | **< 0.01** |

BMI–Body mass index; CCI–Charlson comorbidity index.

## Mortality and hospitalisation rates

There was a non-significant trend to higher mortality in patients with disseminated NTM disease in the time to event analysis (p = 0.24): During the observation period, 8/37 or 29.1% of patients deceased in the case group (disseminated NTM disease) vs. 6/37 or 18.3% in controls.

28/37 (75.7%) patients with disseminated NTM infections were rehospitalised with a median duration to first rehospitalisation of 40.5 days, whereas 21 control patients were rehospitalised (64.9%). The time to event analysis showed that rehospitalisation took place significantly earlier after first admission in disseminated NTM disease (median 40.5 days vs. 109 days, p<0.001) (Fig 3).

A total of 123.61 patient years were observed in disseminated NTM disease and 221.52 patient years in controls. Extrapolation gave 21.77 days of hospitalisation per patient year (PPY) in disseminated NTM disease and 8.75 days PPY in controls. Mean length of stay in hospital was significantly longer in disseminated NTM disease than in controls (21.69 days vs. 10.14 days, p<0.001).

## Treatment and influence on clinical course

In patients with disseminated infection due to NTM, 86.5% received ART. 87.5% of those received regimens containing NRTI, 6.3% NNRTI, 68.8% protease inhibitors, 28.1% integrase

**Table 2. Opportunistic infections, AIDS related malignancies, vegetative symptoms, NTM related diagnostics and complications in patients with disseminated NTM disease and controls.** Opportunistic infections or comorbidities that did neither occur in the case nor in the control group are not shown.

| | Disseminated NTM disease | | | Controls | | | p value |
|---|---|---|---|---|---|---|---|
| | (*n* = 37) | | | (*n* = 37) | | | |
| | n | total | [%] | n | total | [%] | |
| **Non-AIDS defining illnesses (B)** | | | | | | | |
| Chronic diarrhea | 8 | 37 | 21.6 | 6 | 37 | 16.2 | 0.77 |
| ITP | 1 | 37 | 2.7 | 2 | 37 | 5.4 | 1.00 |
| Oropharyngeal or vulvovaginal candidiasis | 26 | 36 | 72.2 | 24 | 37 | 64.9 | 0.62 |
| Herpes zoster | 2 | 37 | 5.4 | 0 | 37 | 0.0 | 0.49 |
| Leucoplakia | 1 | 37 | 2.7 | 0 | 37 | 0.0 | 1.00 |
| **Non-AIDS-defining Malignoma (B)** | | | | | | | |
| Hodgkin-lymphoma | 0 | 37 | 0.0 | 1 | 37 | 2.7 | 1.00 |
| **AIDS-defining illnesses (C)** | | | | | | | |
| Wasting Syndrome | 8 | 37 | 21.6 | 2 | 37 | 5.4 | 0.09 |
| HIV-associated encephalopathy | 1 | 37 | 2.7 | 2 | 37 | 5.4 | 1.00 |
| AIDS defining opportunistic infections | | | | | | | |
| Protozoa | | | | | | | |
| Toxoplasmosis | 6 | 37 | 16.2 | 2 | 37 | 5.4 | 0.26 |
| Cryptosporiodosis | 0 | 37 | 0.0 | 2 | 37 | 5.4 | 0.49 |
| Fungi | | | | | | | |
| PCP | 14 | 37 | 37.8 | 6 | 37 | 16.2 | 0.07 |
| Invasive candidiasis | 10 | 37 | 27.0 | 8 | 37 | 21.6 | 0.79 |
| Cryptococcosis | 1 | 37 | 2.7 | 0 | 37 | 0.0 | 1.00 |
| Histoplasmosis | 1 | 37 | 2.7 | 0 | 37 | 0.0 | 1.00 |
| Bacterial infections | | | | | | | |
| Recurrent bacterial pneumonia | 0 | 37 | 0.0 | 5 | 37 | 13.5 | 0.05 |
| NTM | 37 | 37 | 100 | 0 | 37 | 0.0 | **0.00** |
| TB | 2 | 37 | 5.4 | 6 | 37 | 16.2 | 0.26 |
| *Salmonella* | 1 | 37 | 2.7 | 1 | 37 | 2.7 | 1.00 |
| Viral infections | | | | | | | |
| CMV | 16 | 37 | 43.2 | 4 | 37 | 10.8 | **<0.01** |
| HSV | 12 | 37 | 32.4 | 4 | 37 | 10.8 | 0.05 |
| VZV | 5 | 37 | 13.5 | 0 | 37 | 0.0 | 0.05 |
| PML (Polyoma) | 1 | 37 | 2.7 | 1 | 37 | 2.7 | 1.00 |
| **AIDS-defining malignoma (C)** | | | | | | | |
| Kaposi-Sarcoma | 2 | 37 | 5.4 | 1 | 37 | 2.7 | 1.00 |
| **Vegetative Symptoms** | | | | | | | |
| Fever | 27 | 36 | 75.0 | 12 | 37 | 32.4 | **<0.001** |
| Nightsweat | 13 | 24 | 54.2 | 6 | 37 | 16.2 | **<0.01** |
| Weightloss | 29 | 34 | 85.3 | 10 | 37 | 27.0 | **<0.001** |
| **NTM related diagnostic measures** | | | | | | | |
| Mycobacterial blood culture | 33 | 37 | 89.2 | 17 | 37 | 45.9 | **<0.001** |
| Positive | 26 | 33 | 78.8 | NA | NA | NA | |
| Sputum culture | 25 | 37 | 67.6 | 19 | 37 | 51.4 | 0.24 |
| Positive | 13 | 25 | 52.0 | NA | NA | NA | |
| BAL culture | 19 | 37 | 51.4 | 12 | 37 | 32.4 | 0.16 |
| Positive | 9 | 19 | 47.4 | NA | NA | NA | |
| Other specimen positive for NTM | 28 | 37 | 75.7 | NA | NA | NA | |

(*Continued*)

**Table 2.** (Continued)

| | Disseminated NTM disease | | | Controls | | | p value |
|---|---|---|---|---|---|---|---|
| | (n = 37) | | | (n = 37) | | | |
| | n | total | [%] | n | total | [%] | |
| **Complications** | | | | | | | |
| Documented IRIS | 7 | 37 | 18.9 | 1 | 37 | 2.7 | 0.06 |
| 30-day mortality | 3 | 37 | 8.1 | 0 | 37 | 0.0 | 0.24 |

ITP–idiopathic thrombocytopenia; PNP–polyneuropathy; PCP–*Pneumocystis jirovecii* pneumonia; TB—tuberculosis; CMV–cytoemegalovirus; HSV–Herpes simplex virus; VZV–Varicella zoster virus; IRIS–immune reconstitution syndrome.

inhibitors, and 3.1% a fusion inhibitor (Table 4). Of the five patients that did not receive ART during the observation period, three died before a possible initiation of ART. In controls, 89.2% received ART, of those 96.2% a NRTI containing regimen, 3.1% NNRTI, 75.0% protease inhibitors, 78.1% integrase inhibitors, 0.0% fusion inhibitors and 6.3% CCR5-inhibitors. Integrase inhibitors were prescribed significantly more often in controls (p<0.001).

**Table 3. Other infections, comorbidities and behavioral risk factors in patients with disseminated NTM disease and controls.**

| | Disseminated NTM disease | | | Controls | | | p value |
|---|---|---|---|---|---|---|---|
| | (n = 37) | | | (n = 37) | | | |
| | n | total | [%] | n | total | [%] | |
| **Other infections** | | | | | | | |
| *Aspergillus* spp. | 2 | 37 | 5.4 | 0 | 37 | 0.0 | 0.49 |
| Leishmaniasis | 0 | 37 | 0.0 | 1 | 37 | 2.7 | 1.00 |
| Syphillis | 4 | 37 | 10.8 | 3 | 37 | 8.1 | 1.00 |
| EBV-infection | 1 | 37 | 2.7 | 2 | 37 | 5.4 | 1.00 |
| Hepatitis B | 8 | 37 | 21.6 | 9 | 37 | 24.3 | 1.00 |
| Hepatitis C | 7 | 37 | 18.9 | 9 | 37 | 24.3 | 0.78 |
| **Other Comorbidities** | | | | | | | |
| Diabetes | 3 | 37 | 8.1 | 1 | 36 | 2.8 | 0.61 |
| Liver disease | 12 | 37 | 32.4 | 13 | 37 | 35.1 | 1.00 |
| Chronic kidney disease | 4 | 37 | 10.8 | 2 | 37 | 5.4 | 0.67 |
| Chronic heart failure | 4 | 37 | 10.8 | 1 | 37 | 2.7 | 0.36 |
| Myocardial infarction | 1 | 37 | 2.7 | 1 | 37 | 2.7 | 1.00 |
| Peripheral vascular disease | 2 | 37 | 5.4 | 2 | 37 | 5.4 | 1.00 |
| Cerebral vascular disease | 0 | 37 | 0.0 | 3 | 37 | 8.1 | 0.24 |
| Dementia | 1 | 37 | 2.7 | 1 | 37 | 2.7 | 1.00 |
| Hemiplegia | 0 | 37 | 0.0 | 1 | 37 | 2.7 | 1.00 |
| Peptic ulcer | 2 | 37 | 5.4 | 2 | 37 | 5.4 | 1.00 |
| COPD | 1 | 37 | 2.7 | 3 | 37 | 8.1 | 0.61 |
| Non-AIDS related malignoma | 1 | 37 | 2.7 | 4 | 36 | 11.1 | 0.20 |
| **Behavioral Riskfactors** | | | | | | | |
| Smoking | 13 | 34 | 38.2 | 15 | 36 | 41.7 | 0.81 |
| Alcohol | 7 | 34 | 20.6 | 11 | 35 | 31.4 | 0.41 |
| IVDU | 7 | 37 | 18.9 | 10 | 37 | 27.0 | 0.58 |

EBV—Epstein-Barr Virus; COPD–chronic obstructive pulmonary disease; IVDU–intravenous druguse.

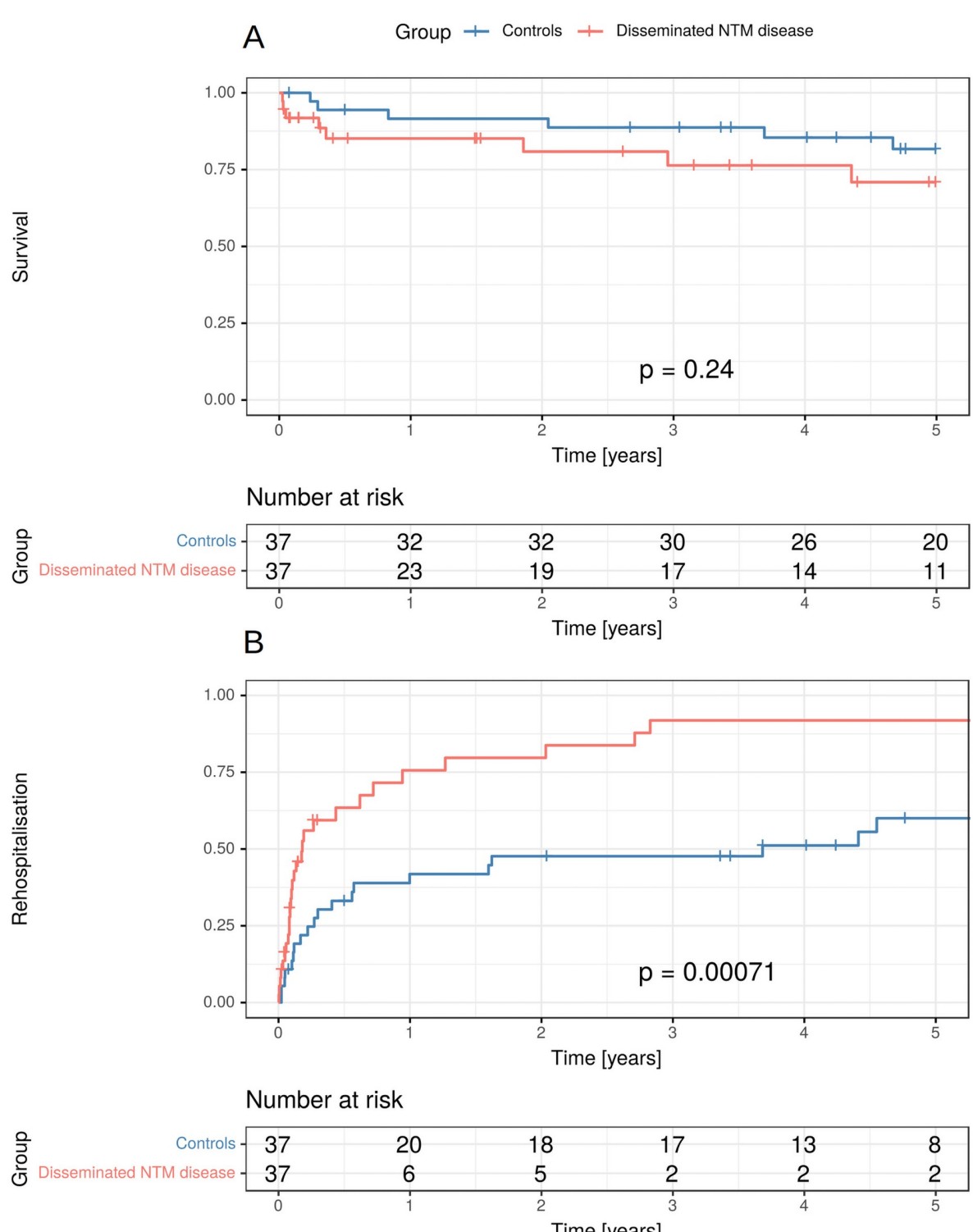

**Fig 3. Survival analysis (A) and time to first rehospitalisation (B) in patients with disseminated NTM disease and controls.** There was a non-significant trend to higher mortality in patients with disseminated NTM disease (p = 0.24). Time to rehospitalisation was significantly shorter in those patients (p<0.001).

**Table 4. Antiretroviral and antimycobacterial treatment in the case group and controls.**

| | Disseminated NTM disease | | | Controls | | | p value |
| | ($n = 37$) | | | ($n = 37$) | | | |
| | n | total | [%] | n | total | [%] | |
|---|---|---|---|---|---|---|---|
| **HIV Treatment** | 32 | 37 | 86.5 | 33 | 37 | 89.2 | 1 |
| NRTI | 28 | 32 | 87.5 | 31 | 33 | 93.9 | 0.43 |
| NNRTI | 2 | 32 | 6.3 | 1 | 33 | 3.0 | 0.61 |
| PI | 22 | 32 | 68.8 | 24 | 33 | 72.7 | 0.79 |
| Integrase inhibitor | 9 | 32 | 28.1 | 25 | 33 | 75.7 | **<0.001** |
| Fusion inhibitor | 1 | 32 | 3.1 | 0 | 33 | 0.0 | 0.49 |
| CCR5-inhibitor | 0 | 32 | 0.0 | 2 | 33 | 6.1 | 0.49 |
| **First NTM treatment** | 29 | 37 | 78.4 | 10 | 37 | 27.0 | **<0.001** |
| Macrolide | 26 | 29 | 89.7 | 3 | 10 | 30.0 | **<0.001** |
| Ethambutol | 27 | 29 | 93.1 | 2 | 10 | 20.0 | **<0.001** |
| Rifamycin | 11 | 29 | 37.9 | 2 | 10 | 20.0 | 0.44 |
| Fluoroquinolones | 1 | 29 | 3.4 | 6 | 10 | 60.0 | **<0.001** |
| Aminoglycosides | 4 | 29 | 13.8 | 0 | 10 | 0.0 | 0.56 |
| **Second NTM treatment** | 15 | 37 | 40.5 | 0 | 37 | 0.0 | **<0.001** |
| Macrolide | 14 | 15 | 93.3 | NA | NA | NA | |
| Ethambutol | 15 | 15 | 100.0 | NA | NA | NA | |
| Rifamycin | 5 | 15 | 33.3 | NA | NA | NA | |
| Fluoroquinolones | 2 | 15 | 13.3 | NA | NA | NA | |
| Aminoglycosides | 2 | 15 | 13.3 | NA | NA | NA | |

NRTI—nucleos(t)ide reverse transcriptase inhibitor; non-nucleos(t)ide reverse transciptase inhibitor; PI—protease inhibitor.

CCR5—CC motif chemokine receptor 5.

Within patients with disseminated NTM infection, 29/37 (78.4%) received antimycobacterial treatment, which consisted mainly of macrolides (89.7% of drug regimens), ethambutol (93.1%), rifamycins (37.9%), aminoglycosides (13.8%) or fluoroquinolones (3.4%). There was a trend to better survival in patients with antimycobacterial treatment within the case group but significance was not reached (p = 0.1). NTM treatment did not postpone rehospitalisation within the case group (p = 0.41).

NTM effective antibiotic treatment was administered significantly more often in patients with disseminated NTM disease than in controls (78.4% vs. 27%, p<0.001). Also, the choice of antimycobacterial agents differed significantly. Five of the ten control patients who received antimycobacterial treatment received a monotherapy with a fluoroquinolone for another indication. No patient within the control group received a second course of NTM-effective treatment.

## Discussion

To the best of our knowledge, this is the first study that has examined the clinical course and outcome of disseminated NTM disease in HIV patients in the ART era using a retrospective case control study design.

As previously described, disseminated NTM disease affects patients with very low CD4 cell counts [1]. This was also true for our case group. Nevertheless, there were no significant differences between groups regarding baseline characteristics, including CD4 cell count, after matching. We therefore compared two groups in a similar immunological state. Fever, night

sweat and weight loss were significantly more frequent in the case group. These simple clinical parameters seem to remain good predictors of mycobacterial disease in these patients.

Interestingly, blood cultures for mycobacterial culture were taken significantly more often in the case group, so that there might be unrecognized cases of disseminated NTM disease in the control group and the statistical effects shown in our study would be slightly underestimated. Additionally, CMV infection was more frequent in those with disseminated NTM disease. Most probably, this is due to the diagnostic work-up in fever of unknown origin that includes mycobacterial cultures and CMV-PCR from the blood.

We could not demonstrate a significant difference in mortality between patients with disseminated NTM disease and controls. This could be linked to low case numbers but underlines the benefit of improvements in the clinical care these patients receive since the introduction of antiretroviral therapy. On the other hand, rehospitalisation took place significantly earlier in disseminated NTM disease than in controls and IRIS was more frequently diagnosed. Taken together, this confirms the clinical observation, that patients suffering from disseminated NTM disease are severely ill, but can achieve convalescence through adequate antimycobacterial and antiretroviral therapy. We therefore argue that those patients require specialised medical attention in an outpatient setting.

Antimycobacterial treatment did not have a significant effect on mortality nor on time to rehospitalisation in disseminated NTM disease. These results should be interpreted with care, as case numbers for time to event analysis within the group of disseminated NTM disease were considerably low. In addition, most patients who did not receive NTM treatment were those who died before its possible initiation. The results do not allow any interpretation regarding current guidelines of NTM prophylaxis in patients with low CD4 counts.

One of the strengths of this study is the relatively large case group which included 37 patients. Using a retrospective case control study design, we compared two groups in similar states of immunosuppression and were therefore able to examine effects attributable to disseminated NTM disease. On the other hand, it has several limitations: First, matching was only possible in a 1:1 manner, as we could not acquire enough controls with equally low CD4 cell counts. Second, this is a monocentric study. Third, a retrospective study design always bears the risk of inaccurate documentation and therefore missing data.

## Conclusion

In this retrospective case control study, we showed that even in the era of highly effective ART, there are differences in clinical course and outcome attributable to disseminated NTM disease in HIV patients. Although mortality was not significantly increased in disseminated NTM disease, we observed significantly earlier rehospitalisation and longer mean hospital stays in those patients. In conclusion, HIV patients with disseminated NTM disease require particularly close monitoring after discharge.

## Author Contributions

**Conceptualization:** Nils Wetzstein, Ari Geil, Timo Wolf.

**Data curation:** Nils Wetzstein, Ari Geil.

**Formal analysis:** Nils Wetzstein, Ari Geil, Annette Lehn.

**Investigation:** Nils Wetzstein, Ari Geil.

**Methodology:** Nils Wetzstein, Annette Lehn, Tobias M. Bingold, Thomas A. Wichelhaus, Timo Wolf.

**Project administration:** Nils Wetzstein.

**Resources:** Nils Wetzstein, Ari Geil, Maria J. G. T. Vehreschild, Timo Wolf.

**Software:** Nils Wetzstein, Ari Geil.

**Supervision:** Nils Wetzstein, Timo Wolf.

**Validation:** Nils Wetzstein, Annette Lehn.

**Visualization:** Nils Wetzstein, Ari Geil.

**Writing – original draft:** Nils Wetzstein, Timo Wolf.

**Writing – review & editing:** Nils Wetzstein, Ari Geil, Gerrit Kann, Annette Lehn, Gundolf Schüttfort, Johanna Kessel, Tobias M. Bingold, Claus P. Küpper-Tetzel, Annette Haberl, Christiana Graf, Maria J. G. T. Vehreschild, Christoph Stephan, Michael Hogardt, Thomas A. Wichelhaus, Timo Wolf.

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
