## [Decision Letter · Decision Letter 0]

17 May 2021

PONE-D-21-03465

Disseminated disease due to non-tuberculous mycobacteria in HIV positive patients is associated with earlier rehospitalisation but not with higher mortality: a retrospective case control study.

PLOS ONE

Dear Dr. Wetzstein,

Thank you for submitting your manuscript to PLOS ONE. After careful consideration, we feel that it has merit but does not fully meet PLOS ONE’s publication criteria as it currently stands. Therefore, we invite you to submit a revised version of the manuscript that addresses the points raised during the review process.

We look forward to receiving your revised manuscript.

Kind regards,

Qigui Yu, M.D./Ph.D

Academic Editor

PLOS ONE

Nancy Beam, PhD

Staff Editor

PLOS ONE

Journal Requirements:

1. In the ethics statement in the manuscript and in the online submission form, please provide additional information about the patient records used in your retrospective study, including: a) whether all data were fully anonymized before you accessed them; b) the date range (month and year) during which patients' medical records were accessed; c) the date range (month and year) during which patients whose medical records were selected for this study sought treatment. If the ethics committee waived the need for informed consent, or patients provided informed written consent to have data from their medical records used in research, please include this information.

" Nils Wetzstein, Ari Geil, Gerrit Kann, Annette Lehn, Johanna Kessel, Tobias M. Bingold, Claus P. Küpper-Tetzel, Christiana Graf and Thomas A. Wichelhaus have nothing to disclose.

Gundolf Schüttfort received funding from Gilead Sciences and speaker fees from ViiV Healthcare, Bristol Meyer Squibb, MSD and Hormosan for participation in Advisory Boards, Data Safety and Monitoring Boards and for preparation of educational materials and lecturing fees, all outside the submitted work.

Annette Haberl received speaker fees from Gilead Sciences, Janssen-Cilag and MSD and also participated in MSD advisory boards. Support of congress participation was provided by Gilead Sciences and Janssen-Cilag.

Maria JGT Vehreschild has received research grants from 3M, Astellas Pharma, Biontech, DaVolterra, Evonik, Gilead Sciences, Glycom, Immunic, MaaT Pharma, Merck/MSD, Organobalance, Seres Therapeutics, Takeda Pharmaceutical, has received speaker fees from Astellas Pharma, Basilea, Gilead Sciences, Merck/MSD, Organobalance, Pfizer and been a consultant to Alb Fils Kliniken GmbH, Arderypharm, Astellas Pharma, DaVolterra, Farmak International Holding GmbH, Ferring, Immunic AG, MaaT Pharma, Merck/MSD, SocraTec R&D GmbH, all outside the submitted work.

Christoph Stephan has received research grants from Gilead Sciences, Janssen, MSD, ViiV-Healthcare and has received fees for scientific advice from Merck/MSD, and Theratechnologies, all outside the submitted work.

Michael Hogardt received Honoraria for consulting (Chiesi GmbH), and lectures (Chiesi GmbH/Horizon Pharma/Thieme Publisher Group), all outside the submitted work.

Timo Wolf received fees for lectures, consultancies and travel grants from: Gilead, Merck Sharp Dome, Janssen, all outside the submitted work."

Please respond by return email with your amended Competing Interests Statement and we will change the online submission form on your behalf.

Additional Editor Comments (if provided):

Reviewers' comments:

Reviewer's Responses to Questions

**Comments to the Author**

1. Is the manuscript technically sound, and do the data support the conclusions?

Reviewer #1: Yes

Reviewer #2: Yes

2. Has the statistical analysis been performed appropriately and rigorously? 

Reviewer #1: Yes

Reviewer #2: Yes

3. Have the authors made all data underlying the findings in their manuscript fully available?

Reviewer #1: Yes

Reviewer #2: Yes

4. Is the manuscript presented in an intelligible fashion and written in standard English?

Reviewer #1: Yes

Reviewer #2: Yes

5. Review Comments to the Author

Reviewer #1: This paper provides some interesting clinic finding of disseminated NTM disease occurring to HIV patients. The results are straightforward. I just had some minor concerns.

1). In figure 2, will there be any statistical significance across different groups?

2). In figure 3A, although P value is not significant, there are indeed some small difference of survival between control and disseminated NTM groups. Perhaps different statistic method(s) can be used? Or would it be possible to include more patient data?

Reviewer #2: This study, by Wetzstein et al., looked to explore the effect of disseminated NTM disease on HIV patient prognosis. The main conclusions found were 1) Earlier rehospitalization and 2) longer duration of hospital stays. The manuscript should be looked over for grammatical errors/ wording prior to publication. For example, the sentence starting “thus” on lines 213-214. Barring this minor concern, the paper is well written and convincing of its’ conclusion.

6. PLOS authors have the option to publish the peer review history of their article (what does this mean?). If published, this will include your full peer review and any attached files.

Reviewer #1: No

Reviewer #2: No

---

## [Author Response · Author response to Decision Letter 0]

19 Jun 2021

Dear Editor, 

we are very thankful for the careful review of our paper, and the positive comments we received. We have revised the manuscript according to the suggestions and remarks made by the reviewers and hope that it will now be acceptable for publication. The review comprises this rebuttal letter, a marked-copy of our manuscript and an unmarked version.

We responded to the Journal requirements as follows:

Journal Requirements:

1. In the ethics statement in the manuscript and in the online submission form, please provide additional information about the patient records used in your retrospective study, including: a) whether all data were fully anonymized before you accessed them; b) the date range (month and year) during which patients' medical records were accessed; c) the date range (month and year) during which patients whose medical records were selected for this study sought treatment. If the ethics committee waived the need for informed consent, or patients provided informed written consent to have data from their medical records used in research, please include this information.

We added the necessary information at the beginning of the Materials and Methods section: 

a) “Before further statistical analysis, all patient’s data were anonymised.” (line 80).

b) “Medical records were accessed between October 2018 and December 2020.” (line 78-79).

c) “For the period between 2006 and 2016, all patients treated at our centre with coded HIV infection (ICD codes B20-B24), coded NTM infection (ICD code A31) and positive microbiological results for NTM were identified and doubles excluded.” (line 74-78)

As this is a retrospective study, informed consent was not necessary. This was added to lines 79-80. 

"Nils Wetzstein, Ari Geil, Gerrit Kann, Annette Lehn, Johanna Kessel, Tobias M. Bingold, Claus P. Küpper-Tetzel, Christiana Graf and Thomas A. Wichelhaus have nothing to disclose.

Gundolf Schüttfort received funding from Gilead Sciences and speaker fees from ViiV Healthcare, Bristol Meyer Squibb, MSD and Hormosan for participation in Advisory Boards, Data Safety and Monitoring Boards and for preparation of educational materials and lecturing fees, all outside the submitted work.

Annette Haberl received speaker fees from Gilead Sciences, Janssen-Cilag and MSD and also participated in MSD advisory boards. Support of congress participation was provided by Gilead Sciences and Janssen-Cilag.

Maria JGT Vehreschild has received research grants from 3M, Astellas Pharma, Biontech, DaVolterra, Evonik, Gilead Sciences, Glycom, Immunic, MaaT Pharma, Merck/MSD, Organobalance, Seres Therapeutics, Takeda Pharmaceutical, has received speaker fees from Astellas Pharma, Basilea, Gilead Sciences, Merck/MSD, Organobalance, Pfizer and been a consultant to Alb Fils Kliniken GmbH, Arderypharm, Astellas Pharma, DaVolterra, Farmak International Holding GmbH, Ferring, Immunic AG, MaaT Pharma, Merck/MSD, SocraTec R&D GmbH, all outside the submitted work.

Christoph Stephan has received research grants from Gilead Sciences, Janssen, MSD, ViiV-Healthcare and has received fees for scientific advice from Merck/MSD, and Theratechnologies, all outside the submitted work.

Michael Hogardt received Honoraria for consulting (Chiesi GmbH), and lectures (Chiesi GmbH/Horizon Pharma/Thieme Publisher Group), all outside the submitted work.

Timo Wolf received fees for lectures, consultancies and travel grants from: Gilead, Merck Sharp Dome, Janssen, all outside the submitted work."

We included the according statement in the competing interests section (line 369)..

Please respond by return email with your amended Competing Interests Statement and we will change the online submission form on your behalf.

This statement was also sent via email to the editorial office prior to this revision.

The ORCID account of the corresponding author, Nils Wetzstein, was linked to the editorial manager with the following iD: 0000-0002-5589-4668.

The reference list was checked to be complete and correct.

Finally, please see attached our remarks to the reviewers’ comments:

Reviewers' comments:

Reviewer #1: 

This paper provides some interesting clinic finding of disseminated NTM disease occurring to HIV patients. The results are straightforward. I just had some minor concerns.

Thank you for your encouraging comments!

1). In figure 2, will there be any statistical significance across different groups?

After matching, there was no statistical difference between the case and the control group as stated in lines 123-124 (p=0.33). CD4 cell counts were significantly higher in the control group before matching (p<0.0001).For clarification, we added these p values to figure 2.

2). In figure 3A, although P value is not significant, there are indeed some small difference of survival between control and disseminated NTM groups. Perhaps different statistic method(s) can be used? Or would it be possible to include more patient data?

You are absolutely right. Mortality was slightly elevated in patients with disseminated NTM disease in comparison to controls (29.1 % vs. 18.3 %, p=0.24, lines 166-168). We cannot use another statistical method as the log-rank test is recommended in this situation. Of course, case numbers could be a problem. Unfortunately, we were not able to include enough controls with low CD4 cell counts. This is one of the limitations of our study as outlined in lines 237-239.

Reviewer #2: 

This study, by Wetzstein et al., looked to explore the effect of disseminated NTM disease on HIV patient prognosis. The main conclusions found were 1) Earlier rehospitalization and 2) longer duration of hospital stays. 

The manuscript should be looked over for grammatical errors/ wording prior to publication. 

We checked the manuscript for grammatical errors and wording to the best of our knowledge. All changes are marked in the version with track changes of the revised manuscript. You can find them in lines 32, 53, 54, 60, 62, 81, 157, 197-198, 205, 209-210, 21, 214-216, 232, 234-235 and 245-246. In addition, we decided to shorten the title in order to make it more concise. It now reads: 

“Disseminated disease due to non-tuberculous mycobacteria in HIV positive patients: a retrospective case control study.”

If any another linguistic adjustments are necessary, we will be happy to provide them.

For example, the sentence starting “thus” on lines 213-214.

We changed this sentence as well (now in lines 214-216): “Interestingly, blood cultures for mycobacterial culture were taken significantly more often in the case group, so that there might be unrecognized cases of disseminated NTM disease in the control group and the statistical effects shown in our study would be slightly underestimated.”

Barring this minor concern, the paper is well written and convincing of its’ conclusion.

If any additional information is required, we are happy to be of assistance.

Best regards, on behalf of the team of authors,

Nils Wetzstein

---

## [Editor Report · Decision Letter 1]

30 Jun 2021

Disseminated disease due to non-tuberculous mycobacteria in HIV positive patients: a retrospective case control study.

PONE-D-21-03465R1

Dear Dr. Wetzstein,

We’re pleased to inform you that your manuscript has been judged scientifically suitable for publication and will be formally accepted for publication once it meets all outstanding technical requirements.

Kind regards,

Qigui Yu, M.D./Ph.D

Academic Editor

PLOS ONE

Additional Editor Comments (optional): minor comments from one reviewer need to be addressed.
---

## [Editor Report · Acceptance letter]

2 Jul 2021

PONE-D-21-03465R1 

Disseminated disease due to non-tuberculous mycobacteria in HIV positive patients: a retrospective case control study. 

Dear Dr. Wetzstein:

I'm pleased to inform you that your manuscript has been deemed suitable for publication in PLOS ONE. Congratulations! Your manuscript is now with our production department. 

Kind regards, 

on behalf of

Dr. Qigui Yu 

Academic Editor

PLOS ONE